# A New Big Data Processing Framework for the Online Roadshow

**Kang-Ren Leow, Meng-Chew Leow *** and **Lee-Yeng Ong ***

Faculty of Information Science and Technology, Multimedia University, Jalan Ayer Keroh Lama,
Melaka 75450, Malaysia; 1132702286@student.mmu.edu.my
*  Correspondence: mcleow@mmu.edu.my (M.-C.L.); lyong@mmu.edu.my (L.-Y.O.)

**Abstract:** The Online Roadshow, a new type of web application, is a digital marketing approach that aims to maximize contactless business engagement. It leverages web computing to conduct interactive game sessions via the internet. As a result, massive amounts of personal data are generated during the engagement process between the audience and the Online Roadshow (e.g., gameplay data and clickstream information). The high volume of data collected is valuable for more effective market segmentation in strategic business planning through data-driven processes such as web personalization and trend evaluation. However, the data storage and processing techniques used in conventional data analytic approaches are typically overloaded in such a computing environment. Hence, this paper proposed a new big data processing framework to improve the processing, handling, and storing of these large amounts of data. The proposed framework aims to provide a better dual-mode solution for processing the generated data for the Online Roadshow engagement process in both historical and real-time scenarios. Multiple functional modules, such as the Application Controller, the Message Broker, the Data Processing Module, and the Data Storage Module, were reformulated to provide a more efficient solution that matches the new needs of the Online Roadshow data analytics procedures. Some tests were conducted to compare the performance of the proposed frameworks against existing similar frameworks and verify the performance of the proposed framework in fulfilling the data processing requirements of the Online Roadshow. The experimental results evidenced multiple advantages of the proposed framework for Online Roadshow compared to similar existing big data processing frameworks.

**Keywords:** Online Roadshow; big data processing framework; Apache Spark; Apache Kafka

## 1. Introduction

In this era of digital business, various efforts are being made to maximize the impact of contactless engagement. Business owners are trying to explore potential solutions to engage their customers and foster alternative interaction. The pandemic changed marketing, and marketing techniques shifted from the physical realm to online, resulting in consumers becoming more accepting of online interactions, especially during movement control periods (i.e., lockdown periods), such as the Movement Control Order (MCO) in Malaysia [1] (and many other countries). A lot of them adopted the virtual interaction platforms through web applications such as e-commerce websites and virtual reality applications. The Online Roadshow, a new type of web application, is a digital marketing approach that aims to maximize contactless business engagement by leveraging web computing and to conduct interactive game sessions via the internet [2]. The Online Roadshow generates massive amounts of user preference data during the period of engagement with participants. The varieties of interactive game data involve data on body movement, voice input, and I/O (input/output) interfaces (e.g., keyboard and mouse) [3]. These data can be categorized into two formats: semi-structured and unstructured (e.g., semi-structured game details and unstructured bodily movement coordination information). This makes it more challenging for traditional programming approaches to process the data [4].

Additionally, the high volume and velocity of the data being generated renders traditional manual data processing insufficient [5]. To address these challenges in the Online Roadshow, there is a need to improve the implementation of data processing with an expected workload so that it can be cost-efficient; have high fault tolerance; and reduce data inaccuracy, inconsistency, and noise to provide quality low-latency clustering outcomes for a better user experience and improve the use of the user engagement data. Historical and real-time big data analysis can enhance contactless business engagement by using big data processing frameworks. Furthermore, big data analytics are crucial for improving the accuracy of the business evaluation and real-time tactical decision-making [6–8].

The ever-growing volume of data exponentially increases the complexity of data processing. Consequently, the concept of big data is introduced to handle and process the massive data sets [9]. The conventional data processing approach is overloaded as it is outrun by the fast advancement of many computing infrastructures [10]. Conventional relational database or storage solutions, such as the MySQL database, are no longer suited to handling gigabytes of data or larger amounts of data as they have long processing times and lower levels of fault tolerance regarding long-term-running workloads. Hence, it is difficult to perform big data analysis using the conventional data analytics approach as it is inefficient due to the five Vs of big data: higher volume, velocity, variety, value, and lower veracity [11,12].

Organizations are looking at building and operating data processing components that can handle data volumes that are growing faster than the computer resource requirements. New methods of implementing data processing tasks with an expected workload in a more cost-efficient way, as well as with higher fault tolerance, are required [13]. Traditional programming models, such as the message passing interface (MPI) and application programming interface (API) definition using ad hoc standards [14], are no longer sufficient for the efficient handling of big data [15]. Therefore, by leveraging advanced big data analysis technologies such as Artificial Intelligence (AI) and Machine Learning, which require a massive dataset to produce reliable output [16], improvements can be made to business strategies and the decision-making process. The concept of big data processing has been introduced to handle and process massive datasets, facilitating better connections between the company and the market [17].

In this paper, a big data processing framework is proposed to provide a better approach for the historical and real-time processing of engagement data from Online Roadshows. The framework enhances the data processing capabilities of big data processing beyond that of traditional programming models to ensure consistency and compatibility while also streamlining processing. The proposed framework not only handles the complexities of semi-structured and unstructured engagement data but it also introduces a pre-processing method to transform the data into a unified semi-structured format that complies with the Complex Event Processing (CEP) paradigm. Moreover, the framework incorporates a dual-mode data processing approach that allows for the simultaneous management of real-time and historical data. This enables a comprehensive analysis of the overall trend and personalized engagement characteristics for each participant. Consequently, this means that the framework could help improve the efficiency, accuracy, and responsiveness of the strategic planning and decision-making processes for companies making use of the Online Roadshow.

By processing the audience preference data of the Online Roadshow in real-time, a targeted advertising approach can better convince the target audience of the value of specific products or services that match their needs, achieving higher marketing effectiveness [18,19] and a better user experience. Web personalization can provide a customized web experience based on the past behaviors of customers, allowing advertisers to deliver a more personalized targeted advertisements [20]. This allows advertisers to iterate relevant marketing messages based on the specific behavioral patterns of consumers. This is predicted to be the future of advertising [21,22]. Historical data processing further facilitates

the evaluation of marketing strategies and trends, enabling businesses to optimize the placement of advertisements and achieve higher engagement rates.

This paper also presents a proof of concept by comparing the performance of the proposed framework with existing big data processing frameworks. This comparison demonstrated the effectiveness of the proposed framework in handling high volumes of semi-structured data. This facilitates more cost-efficient and fault-tolerant data processing, effectively reducing data inaccuracies, inconsistencies, and noise.

In the next section (Section 2), the background, terms, and concepts discussed in the Introduction section will be elaborated. The literature related to big data processing frameworks for Online Roadshows, the background of this study, various existing big data processing frameworks, and the functional modules of big data processing are also presented in the next section. The proposed framework will be presented in Section 3, followed by the experimental environment setup and results in Section 4. In Section 5, the experimental results are discussed to provide an in-depth analysis of the proposed framework. Lastly, in Section 6, the work of the present study is summarized, and conclusions are drawn.

## 2. Literature Review

This section starts by giving the background of this research work on Online Roadshow, a new framework for online advertising. After that, the next subsection provides an in-depth review of big data processing frameworks, their associated capabilities, and functional modules. These include the Application Controller, the Message Broker, the Data Processing Module, and the Data Storage approaches. This subsection also relates these modules to the context of big data processing for the Online Roadshow, explaining its benefits regarding real-time marketing.

### 2.1. Online Roadshow

The Online Roadshow is a digital interactive advertising campaign model that was proposed in [2], developed in response to the COVID-19 (Coronavirus disease 2019) pandemic, which forced individuals to remain socially distanced from one another in an attempt to minimize the spread of the disease [23]. The pandemic disrupted conventional business engagement, as business owners could no longer interact with their customers directly. The framework leverages the web computing power and ubiquitous internet connectivity to achieve higher levels of business engagement in virtual ways, using the VARK (Visual, Audio, Reading and Writing, and Kinesthetic) Learning Model to overcome the ineffectiveness of non-face-to-face interaction [24].

The framework allows business owners to integrate their product- or service-specific advertising components dynamically into the roadshow's activities. The VARK Learning Model is embedded in the interactive games through the use of graphical content, audio content, and interactive content to implement audio visual, reading/writing, or kinesthetic modalities. The interactive games attract participants through various dynamic pieces of content, facilitating the potential exploration of audience preference information, such as audience demographic data and clickstream data. The data analytics module collects and processes the data collected in real-time. The module then generates responsive feedback for business owners through various computing techniques, such as classification and clustering, to provide cues on how to improve user experience [20].

The main purpose of implementing the big data processing approach in the Online Roadshow is to reduce the data inaccuracy, inconsistency, and noise to provide a quality low-latency clustering outcome for a better user roadshow experience. This is in line with similar works like the D-Impact, a pre-processing data algorithm that achieves higher clustering quality by removing noise and outliers [25]. In leveraging the real-time big data processing technique on the clustering procedure, low-latency personalization solutions, such as real-time web personalization, are made possible. Instead of recommending general advertising content to the audience, real-time web personalization can provide a more

customized web experience to the individual users based on their past web interactions [20]. Advertising will become more personalized and targeted, meaning that advertisers can iterate highly relevant advertising messages based on the specific behaviors and needs of consumers. This is the future of advertising [21,22]. Companies are now expected to have a better comprehension of current market trends thanks to the implementation of highly accurate targeted advertising, which allows them to persuasively convince their target audience of the value of their products or services [17,18].

In an optimal personalized advertising scenario, an audience is expected to receive dynamic advertising content based on their preferences, deduced from their behavioral data. The use of clustering techniques on the data to discover customer groups can improve customer experience through personalization. For instance, the navigational data extracted from the weblog can be processed into useful information that can be sent through newsletters with highly relevant suggestions to increase audience satisfaction [20]. Research on targeted advertising indicates that it is an effective marketing approach for a multitude of products and services [26,27]. A successful example of this is Facebook, which leverages targeted advertising techniques to allow business owners to place highly specific targeted advertisements in front of different groups and audiences based on their demographical information, such as hobbies, current living area, age, and gender [28]. This leads to consumers receiving advertisements relevant to their needs, potentially boosting their desire to purchase the advertised product or service (purchase intention) [29]. This process can significantly lower marketing costs and is less time-consuming [30]. Another example of this is the experiment conducted in [31], which assessed the efficacy of chocolate product advertisements targeted towards children. The study proved the efficacy of the advertising approach by evidencing how in enhanced purchase intention and built a positive brand reputation.

### 2.2. Existing Real-Time Big Data Processing Frameworks

With the evolution of rapidly growing information technology, conventional data storage approaches, such as the relational database, are no longer sufficient for the processing of large chunks of data, as they have unacceptably long processing times. Thus, in the 1980s, a big data processing framework, as illustrated in Figure 1, was introduced to eliminate the inefficiency of the conventional data processing framework [32–34]. Data storage modules using the distributed data storage approach on distributed file systems provide more effective data balancing and have fault tolerance capabilities with higher I/O bandwidths. They support higher data loads by improving the structure of the framework, helping to store large amounts of data in a more responsive manner [35]. For example, the Apache Kafka approach of the message broker module intends to improve the inefficiency of traditional polling-based communication, facilitating enhancements in data persistence, volume, velocity, and fault tolerance capability [31]. On the other hand, the Apache Spark approach allows for faster data processing compared to the conventional approach, leveraging the MapReducing programming model, which has parallel programming capabilities [36].

Real-time decision-making capabilities fulfil the requirements of time-critical data processing. For instance, the authors of [37] leveraged the decision-making capabilities of a real-time big data processing framework to manufacture preventive maintenance systems. Device data are collected and further processed to evaluate the health conditions of the manufactured devices using Apache Storm. Moreover, Apache Spark was implemented by the authors of [38] to analyze data collected by the second by multiple sensors in an experimental environment to monitor the operating conditions of a turbine syngas compressor, giving the engineers sufficient time to identify the solutions to potential problems.

On the other hand, the authors of [2] explored the possibility of analyzing environmental monitoring data by using the Apache Spark as middleware to reduce the complexity of the underlying platform, easing the integration of the data processing module and the other devices. Additionally, Apache Spark has been implemented as a stability monitor in

power systems to analyze the data generated by sensors during power production [39]. The stability index is calculated in real-time with an expected minimized latency for monitoring purposes to reduce the possibility of issues such as black out and islanding. A real-time thief identification procedure also helps to locate the crime location by closely monitoring the power grids [39].

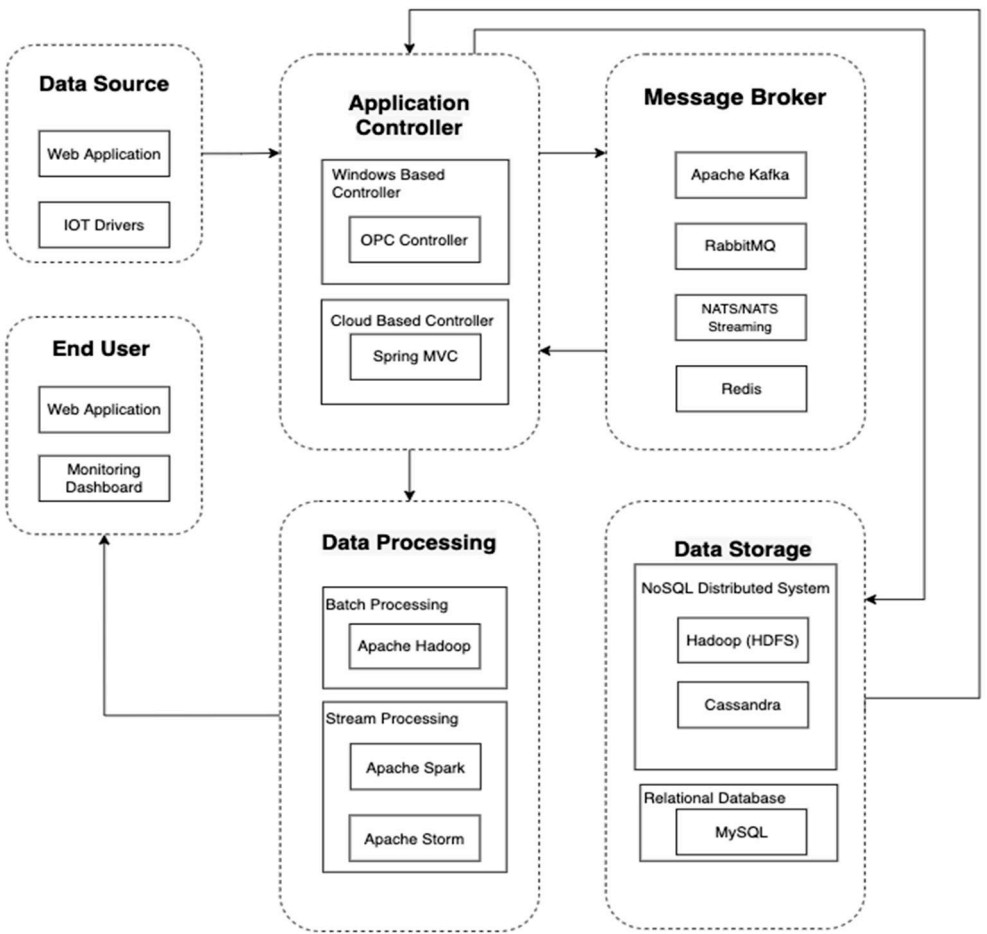

**Figure 1.** The general framework of the big data processing.

Table 1 summarizes the four existing applications of big data processing based on the different modules, as indicated in Figure 1. In order to achieve optimal results for real-time big data processing, several criteria, such as time criticality, data processing mode (e.g., batch and stream processing), hardware specification, etc., have to be considered.

**Table 1.** Existing applications of the big data processing framework.

| Research works | Jabbar, A. et al. (2019) [37]. | Yu, W. et al. (2020) [38]. | Akanbi, A., and Masinde, M. (2020) [2]. | Shyam R. et al. (2015) [39]. |
|---|---|---|---|---|
| **Domain** | Marketing decisions | Manufacturing a predictive alert system | Environment monitoring | Power grid monitoring |
| **Data Diversity** | Transactions (Customer detail, CRM, Orders, Products, IOT, Social Media, Clickstream, Web Data, Search History | Manufacturer device health data (sensor data) | Weather Data | Power Grid Stability, Thief Identification |
| **Application Controller Module** | Cloud based controller | OPC UA | Accessible through Kafka API | N/A |

**Table 1.** *Cont.*

| Data Processing Module | Apache Storm | Apache Spark | Apache Spark | Apache Spark |
|---|---|---|---|---|
| Message Broker Module | N/A | N/A | Apache Kafka | Apache Kafka |
| Data Storage Module | HDFS | Hadoop Distributed File System (HDFS) | N/A | Apache Cassandra |

### 2.3. Application Controller

In web applications, the controller translates user requests or interactions from the view in the Graphic User Interface (GUI) into the operations that the model will perform. Regarding the handling of user interaction requests from the data source, a request handler is responsible for properly processing and handling the request based on the payload [40,41]. The three existing application controller approaches mentioned in Table 1 indicate the important role of the application controller in four different big data processing applications. Regarding Online Roadshows, a cloud-based application controller is necessary as it eases information exchange communication between the client and server. In comparison, the Open Platform Communication (OPC UA) approach used in [38] requires middleware in the web communication process [42]. Two commonly used protocols in communicating and exchanging information between digital devices over a network [43] are taken into consideration for the purpose of message exchange in the cloud-based application controller implementation. They are the SOAP (Simple Object Access Protocol) and REST (Representational State Transfer) protocol.

According to the authors of [44], SOAP is a stateless and one-way message exchange messaging protocol that facilitates communication by using the Hypertext Transfer Protocol (HTTP) and Extensible Markup Language (XML) between SOAP nodes. On the other hand, REST is a client-server architecture that allows clients to send requests for further processing in the server [43]. REST does not limit request parameter to XML, supporting various formats such as JSON (JavaScript Object Notation), strings, etc. With its support for RESTful API (Representational state transfer application program interface) and CEP, Spring MVC's ability to process data without using a proxy service makes it more suitable for use in Online Roadshows. The framework provides better component linking capabilities (e.g., Spark, Kafka, and HDFS) and has a higher-level code usability.

Table 2 shows the comparison between SOAP and REST in terms of coding flexibility and scalability, payload format, payload weight, and bandwidth utilization. These protocols affect the overall operation of big data processing. Leveraging REST protocol, REST API provides an entry point for the exchange of messages. It provides high code flexibility and scalability because there is no need to change the code on the client-side when there are changes in the API. Furthermore, REST API is expected to consume less bandwidth due to its lightweight data format (e.g., JSON-formatted string, etc.).

**Table 2.** Comparison between SOAP and REST.

| Factor | SOAP | REST |
|---|---|---|
| Code flexibility and scalability | Requires changes in client-side code when the service interface changes occur [44]. | Requires no change in client-side code when the REST interface changes [45]. |
| Payload format | Always returns XML data [45]. | Supports various types of data formats [45] (e.g., JSON, Multipart, etc.). |
| Payload weight | Has a heavy payload compared to REST [44,45]. | REST is lightweight as it is meant for the transfer of lightweight data over a commonly known interface (e.g., URI) [44]. |
| Bandwidth consumption | Consumes more bandwidth as SOAP responses may require more than 10 times more bytes compared to REST [44]. | Requires less bandwidth because the response is lightweight (e.g., JSON-formatted string) [45]. |

Thus, the REST protocol is more suitable in the case of Online Roadshow data exchange. The requirement of the CEP paradigm allows for information from the event to flow through IT architecture, makes the data available for processing in real-time prior to storage [46–48]. This provides data transformation or data pre-processing to increase the efficiency of message processing.

### 2.4. Message Broker

A message broker module provides great help in the asynchronous conversation for in-memory data storage processes such as continuous-site following, information examination, IOT data handling, and data logging [49]. The message broker involves two parties: the sender (producer)—who produces the data sent to the message channel—and the receiver (consumer)—who listens to the specific "topic" on the message channel. The four widely used message broker approaches (Apache Kafka, RabbitMQ, NATS/NATS Streaming, and Redis) are listed in Table 3 to summarize their differences in terms of throughput, latency, data persistence, delivery guarantee, and order guarantee [49–51].

**Table 3.** Comparison among existing message broker approaches.

| Author | Bhat, P. J., and D, P. (2020) [49] | | Hegde, R. G., and S, N. G. (2020) [50] | | P. Dobbelaere and K. S. Esmaili. (2017) [52] | | |
|---|---|---|---|---|---|---|---|
| Domain | Messaging System | | Messaging System | | Publish/Subscription System | | |
| Approach | RabbitMQ | NATS/NATS Streaming | Apache Kafka | Apache Kafka | Apache Kafka | RabbitMQ | Redis |
| Throughput | Medium To High | High | High | High | High | Medium To High | Very High |
| Latency | Low-Medium | Higher than RabbitMQ and NATS Streaming | Low | Low | Low | Low-Medium | Very Low |
| Data Persistence | In-memory/Disk | Depends on configuration | OS Cache | OS Cache | OS Cache | In-memory/Disk | In-memory |
| Delivery Guarantee | Yes | Yes | Yes | Yes | Yes | Yes | No |
| Order Guarantee | Yes | Yes | Yes | Yes | Yes | Yes | Yes |

In a typical message broker scenario, a request will be sent to a specific consumer that is awaiting incoming requests from the producers. It consistently handles the message by storing them into an intermediate storage (e.g., disk or memory cache) rather than following the traditional message-processing scenario of directly disposing the message. This eliminates the inefficiency of traditional polling-based communication and eases the message exchange process, increasing the reliability of the web application's request handling of incoming messages (e.g., job request).

Apache Kafka could be introduced as the message broker to properly handle the data. It processes data in the Kafka topic sequentially over the distributed nodes and handles incoming data with high fault-tolerance. Research conducted in [49,50] indicates that Apache Kafka has a higher throughput and lower latency when configured correctly (e.g., appropriate number of topic and replication). Moreover, Apache Kafka keeps messages in the OS cache to offer a message delivery guarantee. Additionally, messages ingested by Apache Kafka will remain in a specific sequence in each topic to guarantee the order of the messages.

On the other hand, RabbitMQ could be introduced as a message-queuing approach for the message broker. It leverages Advanced Message Queuing Protocol (AMQP) to send or receive data from one or more queues, empowering stable and non-concurrent message exchange between applications. RabbitMQ is expected to have medium to high performance in terms of throughput as it relies on acknowledgement (ACK) handshakes and message replication. Message (data) are stored in either memory or disk without steadiness guarantee for message persistence. As RabbitMQ implies optional acknowledge handshakes, it offers delivery guarantee over queue setup [49,50,53].

NATS is an open-source centralized and lightweight message broker that is designed for message exchange between PC applications and administrations. It offers very high throughput performance as it is an elite local application. Moreover, NATS has a higher latency compared to RabbitMQ due to its high idleness. Regarding message persistence, it offers strong storage ability with configurable settings. NATS supports both message delivery guarantee and message order guarantee.

Last but not least, Redis is a message broker that supports in-memory data storage, implementing the publish/subscribe paradigm. It relies on Redis channel, an intermediate storage space, to store published data from the publisher and to allow consumers to consume data from it. Redis has very high throughput performance and a relatively low latency as it stores data in cache memory for rapid access. Redis also supports the configuration of storage, where data persistence can be achieved through in-memory cache or the order of disks, providing a message order guarantee. However, Redis does not offer a message delivery guarantee [50].

Thus, Apache Kafka is considered a better option regarding Online Roadshow implementation. It provides high flexibility in fine-tuning for better performance and integration configuration (e.g., intermediate storage space, message delivery, and sequence guarantee) in big data processing.

### 2.5. Data Processing Approaches

As listed in Table 4, there are two common data processing approaches: batch processing and stream processing (micro-batch processing). The batch processing approach is efficient in processing high volumes of data collected from time to time. On the other hand, the stream processing approach performs data processing with a small window of recent data at one time. This approach can be real-time or near-real-time when there are delays between the time of transaction and changes are propagated [54].

A batch processing approach processes stored data collected from time to time and commonly used Hadoop MapReduce to cluster data into categories, resulting in a longer and more relaxed time interval (e.g., seconds, minutes, or even hours) [10]. The stream data processing, which is a micro-version of the batch processing approach, processes a small window of data immediately and is expected to perform significantly faster than its batch processing counterpart [55]. However, the stream data processing approach (e.g., Spark Stream) is computationally constrained (by resource utilization and configurations, etc.) and is clearly characterized by real-time operation. The stream data processing approach can perform statistical analytics on the fly, which is a particularly important characteristic for streaming data such as user-generated content (in the form of routine user interactions) because the data is arriving continuously at high speed.

**Table 4.** Characteristics of batch and stream processing.

|  | **Batch Processing** | **Stream Processing** |
|---|---|---|
| Data Source [10] | Huge amounts of data being stored in the data warehouse. | Real-time streaming data or micro-batch data. |
| Processing Time [56] | Longer processing times (e.g., seconds, minutes, hours, or days). | Respond interval [57] (e.g., milli-, micro-, or nano seconds). |
| Scenario [56] | Historical data. | System requires low latency (e.g., IOT with sensor, etc.). |
| Examples | Hadoop, Traditional Programming Model. | Apache Spark, Apache Storm. |

Apache Hadoop, Apache Spark, and Apache Storm are three popular data processing approaches. Apache Hadoop was introduced by Google and involves the use of a parallel programming framework [58,59] to perform batch-processing with a Map-Reducing programming model. It provides data storage solutions with distributing capabilities. However, Apache Hadoop can only process stored data and does not support real-time data processing. Therefore, approaches that perform real-time data processing, such as Apache Spark and Apache Storm, have been subsequently introduced.

Apache Spark allows for the implementation of a data processing module in various languages, significantly boosting the flexibility of framework development. It offers a data processing performance that is 10–100 times faster than Apache Hadoop [60], providing more responsive processing results. Aside from its data processing capabilities, Apache

Spark also provides data visualization through Graph X and machine learning capabilities when using the Spark MLib plugin. Thus, it is more functional than Apache Hadoop. Additionally, Apache Spark also supports a variety of data from different file systems, while Apache Hadoop only supports the HDFS. Moreover, Apache Spark is likely to be more suitable than Hadoop in performing iterative tasks due to the latter's performance limitations (for more details, see Aziz et al. [61]).

Apache Storm is introduced by the Apache foundation to support real-time computing. It consists of multiple components that allow for the transfer of data from one data stream to another in a distributed and relatively reliable manner using the directed acyclic graph (DAG) that is topologically ordered. The edges are the data flows, and the vertices are the components. The Spout refers to the data stream source that allows the topology to retrieve data from external data publishers. They are transformed into tuples to emit streams along edges of the directed graph. After that, the processing nodes (called the Bolts) will receive the Spout tuples, consuming input streams to perform further data processing, consequently creating new streams [62–64].

For Online Roadshows, Apache Spark is the most suitable data processing approach, as the Online Roadshow deals mainly with past data (historical data) rather than real-time data streams. Although Spark Stream performs micro-batching rather than actual real-time processing, it is convenient for the Online Roadshow as it can perform data visualization and use machine learning through the plugin available in the Apache Spark environment (e.g., GraphX and SparkML) to better support the interpretation of low-level data analysis.

### 2.6. Data Storage

Relational databases, such as MySQL and Oracle DB [65], are being used in traditional data processing frameworks to store data for effective data centralization and efficient storage management [65]. However, as the huge amount of data goes beyond a certain size, the capabilities of the relational database in collecting, storing, and analyzing the data in the traditional structure become inefficient [66]. It is no longer sufficient for processing the huge amount of data, and the data integrity in the relational manner is compromised [65]. Therefore, Google proposed the Apache Hadoop (HBase) to parallelize the data processing capability of big data by distributing the load to accelerate the computation and reduce latency inefficiencies [67].

A distributed file system with HDFS is a flexible, adaptive, and clustered method of managing the data files in a big data environment with a NoSQL solution. It is a data service that offers the unique set of capabilities required for handling large volumes of data with a high velocity [64]. In HDFS, files and directories are represented as the NameNodes. It records the metadata attributes, such as access permission, modification details, namespace, access histories, and disk space quota, acting as the first contact for allocating the data blocks containing the file. It will then read the respective blocks from the DataNode closest to the client [68]. On the other hand, the DataNode that stores the data sends heartbeats to the NameNode to ensure the DataNode is alive, and the replicated data is available over the cluster [68]. Data Nodes are replicated over multiple nodes and accessed under the control of the NameNode to ensure data integrity over the cluster and prevent file corruption during failure [67].

On the other hand, Apache Cassandra offers high scalability, availability, and fault tolerance through its replicational characteristics. It is designed to handle various types of unstructured, structured, and semi-structured data. Apache Cassandra provides a relational data storage design to allow key value storage, similar to the SQL relational database with foreign key joins. In a Cassandra cluster, nodes are interconnected, and data are distributed and communicated across each independent node in the cluster regardless of its location. The data model of Cassandra is significantly distinct from other relational database management models due to the fact that it supports key values stored in the storage space [69].

In the case of Online Roadshow implementation, a distributed file system is more reliable since the data does not require key value storage and is expected to have higher efficiency in data reading latency and data writing capability. Research conducted by Jakkula et al. [69] indicates that Apache Cassandra can achieve slightly better performance in the operation of updates involving small amounts of data; however, this results in a higher latency when the amount of data increases significantly.

In this section, a comprehensive review of the big data processing framework for Online Roadshows was discussed, covering the functional modules and associated capabilities. With these in mind, the next section presents the proposed framework in detail.

## 3. Proposed Framework

The main purpose of developing the proposed framework is to facilitate personalized participant engagement in the Online Roadshow using dual-mode big data processing. The proposed framework, as illustrated in Figure 2 consists of four mains components: The Application Controller module, the Message Broker module, the Big Data Processing module, and the Data Storage module. These modules are designed to improve the processing of participant engagement data for the Online Roadshow.

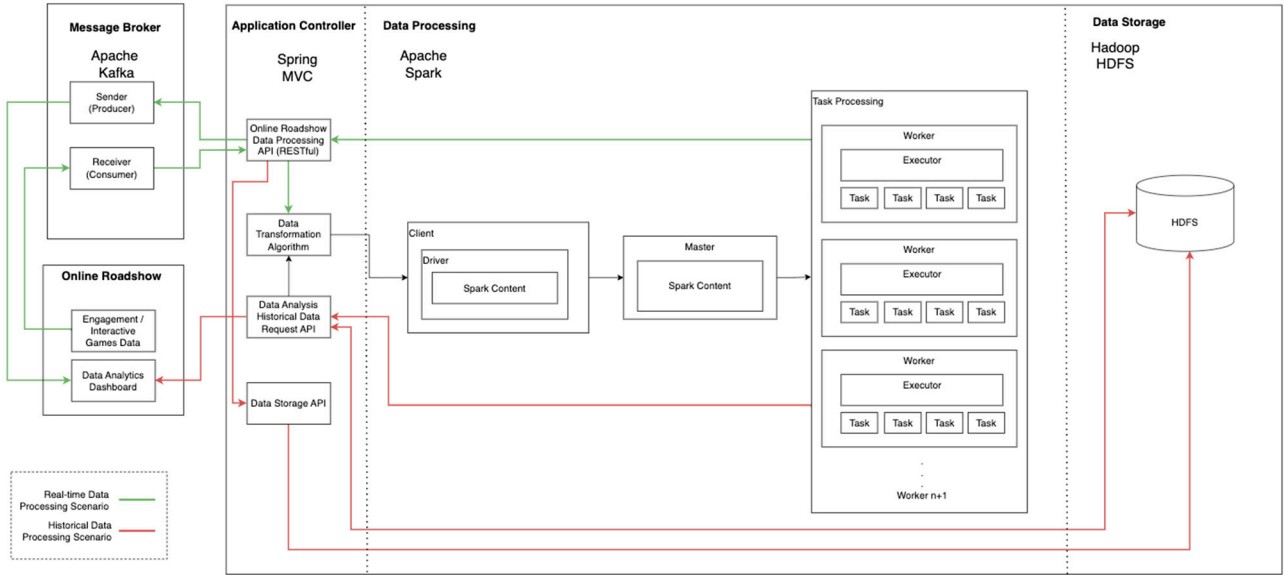

**Figure 2.** Proposed big data processing framework.

A big data processing framework for the Online Roadshow should be capable of working in dual-mode (red line is the Historical Data Processing scenario and green line is the Real-time Data Processing scenario) to analyze responsive (real-time) and batch (historical) data concurrently. It needs to serve different purposes that require faster and slower responses. For example, real-time big data processing provides a more accurate personalized targeted advertising experience, while historical data processing provides highly efficient trend analyses. For real-time data processing, the proposed framework leverages Apache Kafka as the message broker to handle the large amount of incoming data on the participant's preference. Apache Spark is used as the data processor.

By implementing the real-time data processing, web personalization can be used to deliver a customized web experience based on the past user behaviors [20], allowing advertisers to deliver more personalized and targeted advertisements. This practice gives a great opportunity for advertisers to iterate messages and disseminate content that is in alignment with the customer's tastes [21,22]. Furthermore, the historical data processing capabilities of the proposed framework provide quantitative feedback for the Online Roadshow event, offering a clearer understanding of the effectiveness of advertisements and the efficiency of the overall marketing strategy. For instance, analyzing the historical

data helps businesses to identify strategic advertisement spaces and content that may generate higher traffic flow.

Considering the CEP, the proposed framework leverages Spring MVC as its Application Controller module, which provides high-level big data processing system manipulation. It is highly important to improve its system capabilities to process a variety of data types (e.g., JSON String, files, etc.) and for its data processing manipulation. At the same time, it also provides module integration capabilities (e.g., approach configuration, loggings, and multiple methods of data feeding), making the connection between the models implemented in the proposed framework highly flexible and providing API with REST protocol, which supports a variety of data-over-client-server message communication. In order to ingest, process, store, and retrieve the data produced during the Online Roadshow event, four models, consisting of the Online Roadshow Data Processing API, Data Analysis Historical Data Request API, Data Transformation algorithm, and Data Storage API are incorporated into the Application Controller module.

In the Online Roadshow, web engagement and interactive game data such as page visit, gameplay preferences, and gameplay performance data are collected during participant interaction sessions. Thus, the Online Roadshow Data Processing API provides data feed ingestion from the Online Roadshow to the Message Broker module for real-time data input and further processing. The pseudocode of the Online Roadshow Data Processing API is shown in Figure 3. It consumes the data from the message broker and validates the data to ensure a valid semi-structured JSON data format before processing them via the Data Transformation Algorithm, as shown in Figure 4, to transform the data into a key-value pair for the subsequent data extraction process. Invalid data will be ignored and the next record will be processed instead. Then, the pre-processed data will be processed by the Data Processing module and further broadcast to the message broking channel, which allows subscribers in the message broker to retrieve the real-time data processing result while storing the data in the Data Storage module in the background.

---

Online Roadshow Data Processing API

---

Input: Raw data produced by Online Roadshow / Message Broker feed
Output: Results broadcast through Message Broker

---

1: Data Input from API / Message Broker

2: Data pre-processing

3:  forEach(data list)

4:  if (isValidStructure)

5:   Pre-processing (Data transformation Algorithm in Figure 5)

6:  end loop

7:  Return result

---

**Figure 3.** Online Roadshow Data Processing API pseudocode.

**Data Transformation Algorithm**

**Input: Raw data received by Online Roadshow Data Processing API**
**Output: Formatted string for Data Processing module**

1: Data Input from Online Roadshow Data Processing API

2: Extract campaign id, game id and time from data

3: if (valid time format && game data exists)

4:   Extract detail game data (e.g. Clicks/tried times before game over)

5:   Re-arrange format into key-value pair

     (e.g. campaign_id;game_id;time;game_data)

6:   else

     show error message

7:   return result

**Figure 4.** Data Transformation Algorithm pseudocode.

At the same time, historical data ingested from the Message Broker will be stored into the Data Storage module through the Data Storage API, as shown in Figure 5. By analyzing the historical data, advertisers can enhance decision-making efficiency by fine-tuning advertising strategies in the Online Roadshow to facilitate better engagement through providing a personalized web experience. In the case of the historical data retrieval, the Data Analysis Historical Data Request API, as shown in Figure 6, will retrieve the data within the specific requested time frame from the Data Storage module to analyze the overall participant preferences in terms of the sequence of page visit and gameplay behavioral characteristics. After that, the data retrieved will be pre-processed by the Data Transformation Algorithm for further processing by the big data processing module. Finally, the processed data will be returned through the Data Analysis Historical Data Request API to the Data Analytics Dashboard in the Online Roadshow for visualization.

**Data Storage API**

**Input: Raw data produced by Online Roadshow / Message Broker feed**
**Output: Storage operation status (true/ false)**

1: Read request parameter from request body

2:   forEach(data list)

3:   if (isNotNull)

4:     store raw data into Data Storage Module

5: end loop

6: return result

**Figure 5.** Data Storage API pseudocode.

**Data Analysis Historical Data Request API**

**Input: Campaign id, time range, game id (optional)**
**Output: Results return through API**

1: Read request parameter from request body

2: if (valid date)

3:   Retrieve data from Data Storage Module

4:   forEach row of data

5:   if (isValidStructure)

6:     Pre-processing (Data transformation)

7:     build pre-processed historical data array

8: end loop

9: return result

**Figure 6.** Data Analysis Historical Data Request API pseudocode.

The Message Broker module in the proposed framework leverages the Apache Kafka as the request handler. Apache Kafka is used to construct the real-time data streaming pipelines that function as a reliable data handling stream of records from the client applications, such as the sequence of page visit and the game play behavioral characteristics.

There will be two models in the Request Handler module that process the real-time data processing requests. The Receiver (Consumer) model is the data entry point and the Sender (Producer) model returns the processed result.

As a result, the proposed framework can process multiple streams of data on the background in the multi-threaded manner without interrupting the user processes, effectively enhancing the user experience.

The Data Processing module intends to analyze the audience preference data in a more responsive manner to allow the system to achieve higher accuracy in the targeted content of the dynamic advertising of the Online Roadshow implementation. For instance, a responsive analysis increases the efficiency of the product/service promotion because it can deliver preferable content to the audience, potentially influencing the consumer's idea of brand preference and their intention to purchase the product or service. The Data Processing module in the proposed framework implements the Apache Spark with a MapReducing approach. Figure 2 illustrates a Spark Cluster that allows the spark application to run sets of processes independently. For resource optimization, the Spark Cluster can connect to various types of resource managers to optimize the resource allocation process.

During an Online Roadshow event, personal preference data, such as user's page visits, gameplay behavioral characteristics, and gameplay performance, are collected accordingly for each participant. Thus, the Data Storage module is able to store and retrieve the data of a high volume and velocity while maintaining the data integrity, data persistence, and fault tolerance capability, lowering the risk of losing valuable participant information.

## 4. Experiment and Results

### 4.1. Experiment Setup

As discussed in the previous section, the proposed framework aims to provide a personalized participant Online Roadshow engagement by utilizing a dual-mode big data processing module. In this section, the experiments we performed and their corresponding results will be presented. The experiments involved simulating the real-time (via the Message Broker module) and historical data (via the Data Storage module) processing scenarios. This section presents the experimental evaluation of the proposed framework in terms of throughput, latency, and scalability. This section also provides a detailed analysis

of the experimental results, highlighting the strengths and weaknesses of the proposed framework and identifying areas for future improvement.

Specifically, this study aims to evaluate the performance of the proposed framework in comparison to 10 existing combinations of big data processing approaches, as listed in Table 5. Among them, combination #1 to #3 simulated the Historical Data Processing scenarios, where data are retrieved from Data Storage module and processed in batches within the specific time range. Apache HDFS and Apache Cassandra were implemented as the Data Storage module. Data were retrieved from the Data Storage module in the different throughputs and passed to the Data Processing module (e.g., Apache Spark and Apache Storm).

On the other hand, combination #4 to #10 simulated the Real-time Data Processing scenario, where data are ingested and processed within the Data Processing module through the message broker to the message subscribers. Four message brokers (e.g., Redis, RabbitMQ, Kafka, and NATS) were leveraged as the data source. Data were ingested from the Message Broker module and processed by the Big Data Processing module (e.g., Apache Spark and Apache Storm) accordingly with specific data throughput broadcasted via the Message Broker module for message topic/channel subscribers in real-time. Kafka API was not used in the experiment as it does serve the RESTful API.

The experiments were conducted on a standalone device with Intel Core i5-8250U @1.6 Ghz and 12 GB 2666 Mhz DDR4 memory running on Windows 10. The average result of the commonly used metrics (listed in Table 6) were computed for each combination. These evaluation metrices include the execution time, memory usage, CPU usage, and throughput. These measures assessed the performance of the data processing mechanism and their resource consumption requirements.

**Table 5.** Combinations of existing big data processing frameworks.

| Module | Scenario | | Application Controller Module | Message Broker Module | | | | Data Processing Module | | Storage Module | |
|---|---|---|---|---|---|---|---|---|---|---|---|
| Experiments | Real-Time | Historical | Spring MVC | Kafka | RabbitMQ | Redis | NAT/NATS | Spark | Storm | HDFS | Cassandra |
| #1 | | ✓ | ✓ | | | | | ✓ | | ✓ | |
| #2 | | ✓ | ✓ | | | | | | ✓ | ✓ | |
| #3 | | ✓ | ✓ | | | | | | ✓ | | ✓ |
| #4 | ✓ | | ✓ | | ✓ | | | ✓ | | | |
| #5 | ✓ | | ✓ | | | ✓ | | ✓ | | | |
| #6 | ✓ | | ✓ | | | | ✓ | ✓ | | | |
| #7 | ✓ | | ✓ | ✓ | | | | | ✓ | | |
| #8 | ✓ | | ✓ | | ✓ | | | | ✓ | | |
| #9 | ✓ | | ✓ | | | | ✓ | | ✓ | | |
| #10 | ✓ | | ✓ | | | | ✓ | | ✓ | | |

**Table 6.** Commonly used performance metrices.

| Metrices | Description | Formula |
|---|---|---|
| Execution time [61,70–72] | - A period in which an event is actively operating. | Start time of request–End time of request. |
| Memory Usage [70,72–75] | - Memory consumption during task execution. | N/A |
| CPU Usage [61,70,74–76] | - The identical CPU performance scale. <br> - The system spent in different modes of the execution since boot. <br> - How much CPU time is used per minute to process a job in percent. | N/A |
| Throughput [61,70,77,78] | - Units of information that can be processed in a given amount of time. | $T = \frac{n}{time}$ <br> Where: <br> $n$ = number of nodes running <br> $time$ = predefined period (typically seconds) <br> Or <br> $T = \frac{Total\ Byte\ process}{Test\ execution\ time}$ |

Data were collected for the Online Roadshow in two sessions between 5 July 2021 to 31 August 2021 and 12 July 2022 to 21 July 2022, with a total of 504 participants. A

total of 1.97 Gb of Online Roadshow records comprising the participant information, timestamps, gameplay details (e.g., accuracy of the kinesthetic-oriented gameplay and words being used in the audio-oriented game that would help to identify one's personal preferences) was collected in both campaigns of the Online Roadshow. A snippet of the dataset used in the experiment is illustrated in Figure 7, containing 12 fields that provide a comprehensive view of the game results within the Online Roadshow. These fields include Game_Result_ID, Result, Game, Result_create_time, Result_update_time, Campaign_ID, Activity_ID, Participant_ID, User_ID, game_data, device, and type.

| Game_Result_ID | Result | Game | Result_create_time | Result_update_time | Campaign_ID | Activity_ID | Participant_ID | User_ID | game_data | device | type |
|---|---|---|---|---|---|---|---|---|---|---|---|
| 6159 | 200 | null | "2021-06-28 02:30:16" | null | "182" | "5" | null | 633 | {"gameSetting":{"totalsec":120,"commands":{"directions":{"up":["wash"],"down":["vaccination","vaccinations"],"right":["kill"],"left":["home"]},"restart":"restart,"....... | "desktop" | 1 |
| 6177 | 499 | null | "2021-06-28 10:17:28" | null | "182" | "2" | null | 706 | {"gameSetting":{"numOfDogs":6,"numOfCats":1,"numOfHint":5,"totalSec":60},"gamePlay":{"spawnStatus":[{"type":"dog","isTouch":true,"x":319.852482306073,"y":361.9843781308917,"radius":17.2,"....... | "desktop" | 1 |
| 6178 | 260 | null | "2021-06-28 10:17:30" | null | "182" | "6" | null | 712 | {"setting":{"row":7,"column":6,"speed":6,"life":0},"result":{"score":260},"round":[{"level":1,"score":260,"speed":6,"start_time":"2021-06-28 10:17:09","end_time":"2021-06-28 10:17:30"}]} | "desktop" | 1 |

**Figure 7.** Snippet of Online Roadshow engagement data before pre-processing.

The Game_Result_ID serves as a distinctive identifier for each game session, ensuring easy tracking and identification. The Result field denotes the game score achieved during the session, quantifying the participant's performance. The Game field, currently reserved for future utilization, holds potential for additional game-related information. Timestamps are captured through the Result_create_time field, indicating the precise moment of recording the game session result. The Result_update_time, currently reserved for administrative purposes within an associated portal, remains slated for future implementation. To establish a connection with the Online Roadshow campaign, the Campaign_ID field associates each game session with a specific campaign. The Activity_ID field signifies the type of game played during the session, enabling categorization and differentiation. Participant-related information is captured through the Participant_ID and User_ID fields. Participant_ID represents the user name or identifier, while User_ID provides a unique identifier for individual tracking. The game_data field holds intricate details of the game session, encompassing a blend of unstructured and semi-structured data. It includes components such as body coordination matrices, game performance metrics, and play attempts, providing comprehensive insights into participant gameplay behavior and performance. Additional fields in the dataset include device—which indicates the type of device used during the game session (e.g., desktop or mobile)—and type—a reserved field intended for potential future use.

During the experiments, the message broker acted as the entry point to access and trigger a processing request in the proposed framework. Data collected in the Online Roadshow were fed into the message broker programs (e.g., Kafka, Redis, RabbitMQ, and NAT) in different volumes, ranging from 100 to 2567 records, to simulate the scenario of real-time data processing. On the other hand, data were retrieved from the Data Storage

module (e.g., HDFS and Cassandra) to simulate the Historical Data Processing scenario. The data ingested were then processed by the data processing module (e.g., Spark and Storm). Since Apache Storm does not support batch processing, the Data Storage module was used as the data source (instead of the Message Broker module) for the Historical Data Processing scenario.

To accurately evaluate the performance of the proposed framework and the other combinations, it is essential to use reliable tools to capture and monitor the usage data. In this paper, Glowroot, a Java Virtual Machine (JVM) performance monitoring tool, was utilized to capture the resource consumption and task execution data in a 5-s interval. This allowed for a detailed review of the memory and CPU usage during task execution. The captured data were then used to generate comprehensive performance comparison charts among the big data processing framework combinations, with CPU usage, execution time, and memory usage being among the key measures captured. The results of the performance evaluation are presented in the next section.

*4.2. Result*

In this section, we present the results from our experiment to help facilitate a comparison between the combinations of the existing big data processing frameworks and the proposed framework. The average time and resource consumption of the execution of different throughputs from 100 to 2567 records for the two different scenarios were measured in each experiment. The combinations that leveraged the Data Storage module were categorized into the Historical Data Processing scenario, while the combinations that involved the Message Broker module were categorized into the Real-time Data Processing scenario.

4.2.1. Historical Data Processing Scenario

Figure 8 shows a comparison of the execution times for the proposed framework and combinations #1, #2, and #3. The lower execution time indicates that this specific combination can process the historical data from the Data Storage module quicker that the others. Among the four frameworks shown in Figure 8, combination #2 (gray triangle) presents the lowest time consumption (0.1468 s) when the throughput is 1500 and below. However, the proposed framework (blue diamond) overtakes combination #2 when the throughput is higher than 1500. The performance of the remaining combinations #1, #2, and #3 shows insignificant changes in execution time when the throughput is lower than 1000, but significant changes occur when the throughput reaches 1000 and above.

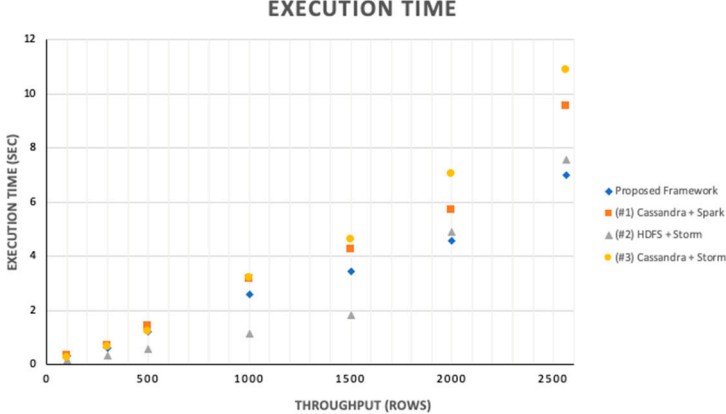

**Figure 8.** Execution time comparison between combinations #1, #2, and #3 and the four frameworks.

Figure 9 presents the results of the performance evaluation for the proposed framework and the other combinations in terms of throughput and CPU usage. The lower CPU consumption indicates better resource utilization, and the proposed framework (blue diamond) demonstrates moderate CPU usage compared to the combinations (ranging from

2.4% to 25.6%). Combination #2 (gray triangle) has the lowest CPU usage most of the time, but when a higher data throughput of 2000 is tested, the proposed framework shows the lowest CPU usage overall (18%). However, the proposed framework is found to have higher CPU usage at a throughput of 2567 (25.6%). In contrast, combination #1 (orange box) consumes the highest CPU usage most of the time. At a lower throughput of below 1000, there is no significant difference between the proposed framework and combinations #1 and #3, but the difference becomes more noticeable at higher throughput levels (e.g., 1000, 1500, 2000, and 2567).

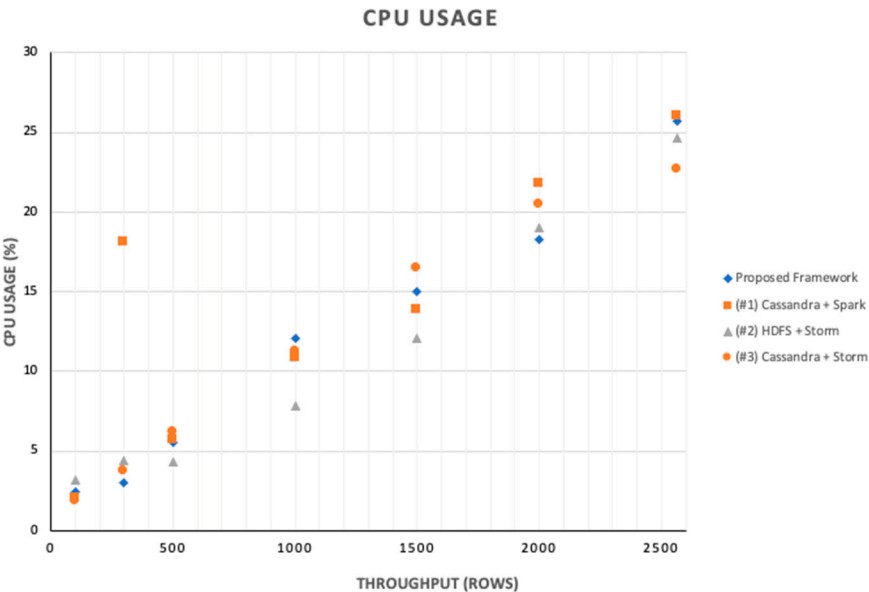

**Figure 9.** CPU usage comparison between combinations #1, #2, and #3 and the four frameworks.

Subsequently, the memory utilization of the proposed framework and the other combinations are compared in terms of their throughput. As shown in Figure 10, the proposed framework demonstrates moderate memory consumption, indicating better efficiency in memory utilization. Combination #3 is found to have almost similar memory consumption to the proposed framework at certain throughputs, while combination #1 shows the lowest memory consumption at lower throughputs but consumes the highest amount of memory at higher throughputs.

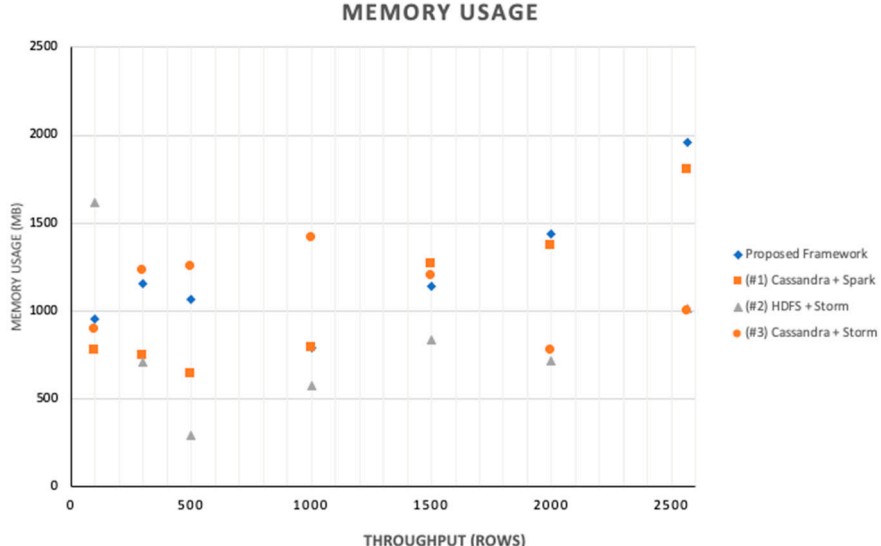

**Figure 10.** Memory usage comparison between combinations #1, #2, and #3 and the four frameworks.

4.2.2. Real-Time Data Processing Scenario

Figure 11 illustrates a comparison regarding execution time for the Real-time Data Processing scenario. A lower execution time indicates that the specific combination can process data quickly before broadcasting the information into the Message Broker module to its subscribers. It can be observed that the proposed framework has the shortest execution time most of the time, except at lower throughputs of 100, 300, and 500 (0.63 s to 6.69 s). However, combination #5 (gray triangle, from 1.06 s to 28.293 s) and #9 (blue plus, 1.64 s to 7.65 s) have the longest and the second longest execution time most of the time (for all throughputs). Aside from the proposed framework, combinations #5 and #9 showed a less significant difference in terms of time consumption compared to combinations #1, #2, #3, #4, #6, #7, #8, and #10.

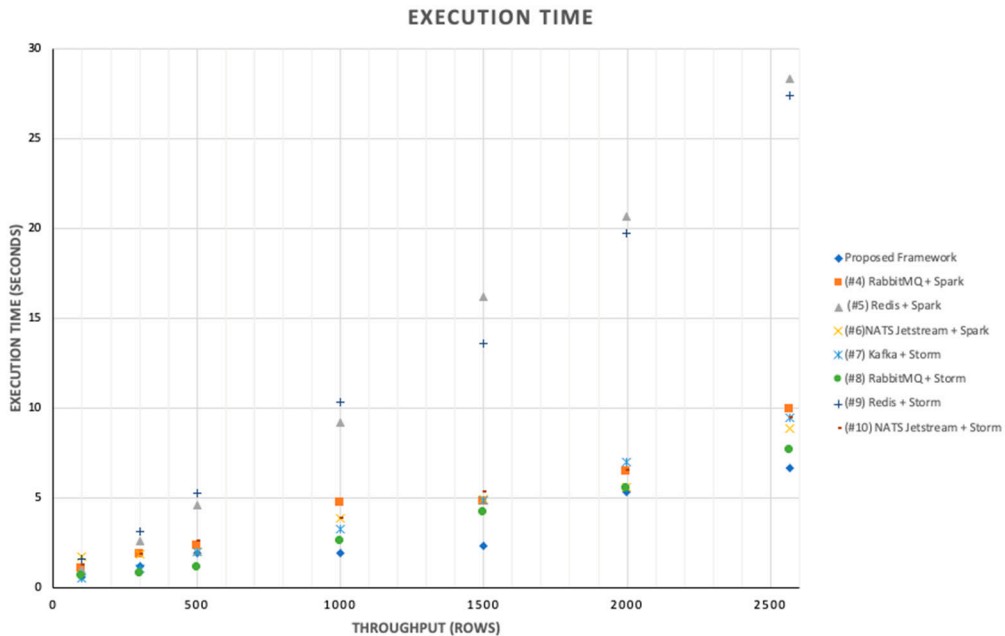

**Figure 11.** Execution time comparison between combinations #4 to #10 and the proposed framework.

Figure 12 illustrates the relationship between throughput and CPU usage (in percentage) for the eight real-time data processing approaches. The proposed framework has the second lowest CPU usage most of the time, and the lowest (10.15%) at the highest throughput (2567). On the other hand, combination #4 (orange box) has the highest CPU usage most of the time (e.g., at throughputs of 100, CPU usage is 4.9%, 300—7.9%, and 1000—18.64%). This is followed by combination #7 (blue star), which has the second highest CPU usage most of the time (e.g., at throughputs of 1000, 1500, and 2000) and has the highest CPU usage at the highest throughput (2567). CPU usage starts to show significant changes when the throughput hits 500 and above.

Figure 13 illustrates the relationship between throughput and memory usage, measured in Mega Bytes (MB), for the eight real-time data processing approaches. Most of the time and when throughput is higher than 1000, the proposed framework has the lowest memory (from 664 MB to 958 MB). However, combination #10 (red dash) has highest memory usage among the combinations (1951 MB to 2403 MB) almost all of the time.

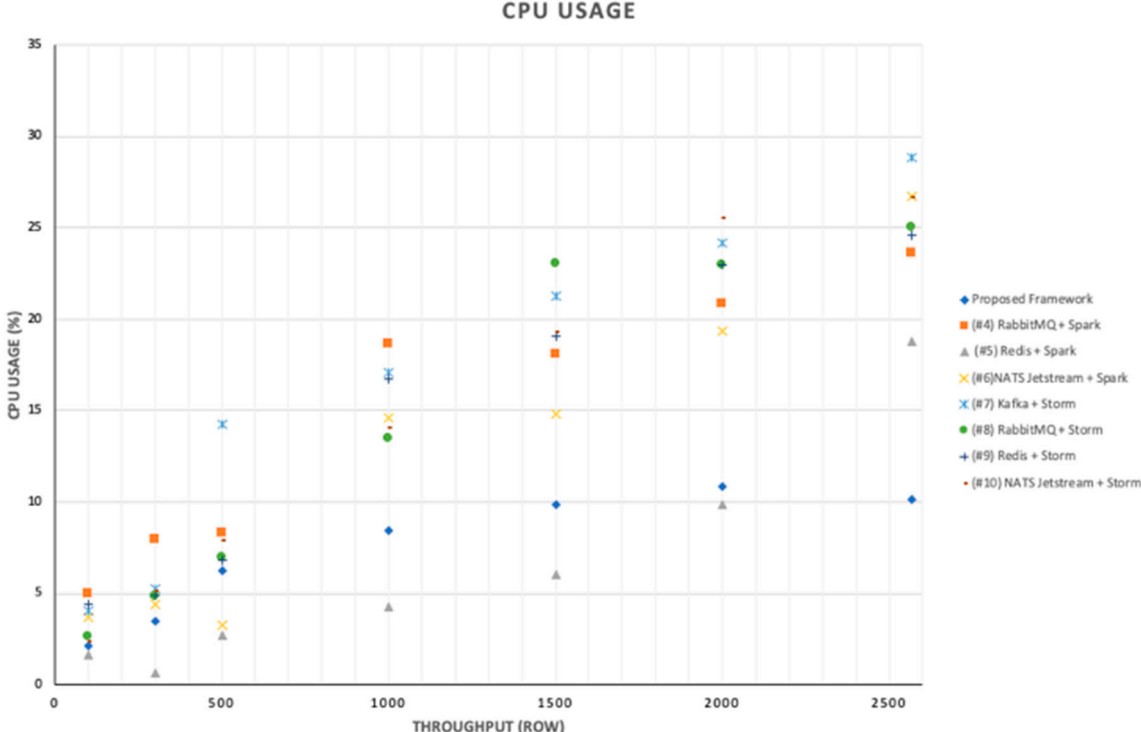

**Figure 12.** CPU usage comparison between combinations #4 to #10 and the proposed framework.

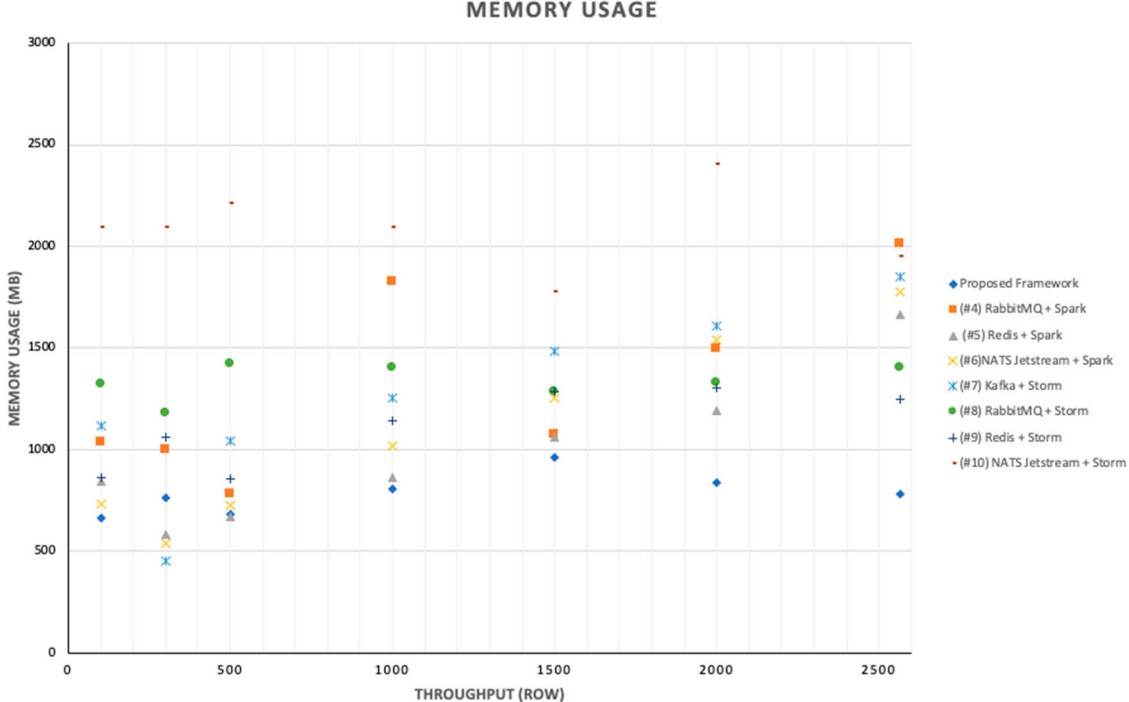

**Figure 13.** Memory usage comparison between combinations #4 to #10 and the proposed framework.

## 5. Discussion

After evaluating the performance of the Historical Data Processing scenario, it is clear that the proposed framework outperforms all existing data processing frameworks in terms of execution time. In the CPU usage, the proposed framework shows moderate CPU usage among all the combinations when the throughput is low. However, it consumes the second

and third highest memory among all other combinations, and the highest at a throughput of 2567.

On the other hand, the experimental results clearly demonstrate that the proposed framework has the potential to provide a highly efficient and effective solution for real-time data processing in web personalization. The combination of Apache Kafka and Apache Spark has proven to be highly effective in reducing latency and improving performance, resulting in better resource utilization in terms of CPU and memory usage. The ability to handle multiple streams of data in parallel processing has made the proposed framework highly responsive and capable of providing a more personalized web experience. The caching of persisted data in memory also improves the efficiency of the analysis, allowing the framework to perform more efficiently when more data is being fed into the system. Overall, the proposed framework shows great promise for enabling personalized advertising and targeted product recommendations, which can greatly enhance the user experience and improve customer satisfaction.

Although the proposed framework did not perform as well as expected in all aspects during the Historical Data Processing scenario, it demonstrated outstanding performance in terms of execution time, outperforming all other combinations. This can be attributed to the utilization of Apache Spark's Resilient Distributed Datasets (RDD), which leverages more CPU and memory usage to cache a large amount of data in a distributed manner, resulting in higher efficiency in the Map-Reduce operation.

On the other hand, the proposed framework performs the best among all other combinations in the Real-time Data Processing scenario. Although Apache Storm is commonly used for real-time data processing, it may not be suitable for batch processing scenarios since it cannot identify the final tuple in a queue or indicate the range of data indices for further processing. Hence, these findings highlight the significance of the proposed framework in supporting dual-mode data processing (both real-time and historical data) for optimal performance.

In an Online Roadshow event, the dual-mode processing capability provides concurrent real-time and historical data processing. This allows the proposed framework to process a massive volume of game page visits, gameplay behavioral characteristics, and gameplay performances for each participant. The real-time data processing capability is essential for providing responsive feedback, such as web personalization, while the Data Storage module is capable of storing and retrieving data of a high volume and velocity while maintaining the data integrity, data persistence, and fault tolerance capability, lowering the risk of losing valuable participant information.

At the same time, the historical data processing capability of the proposed framework analyzes the overall trend of the advertisement, leading to a higher attention rate, which consequently leads to a higher engagement level. With the aforementioned enhanced data processing capabilities, a truly customer-centered Online Roadshow can be realized, revolutionizing digital advertising practices.

## 6. Conclusions

The ever-growing volume of data production in this present era of technology has led to massive amounts of data being generated. Providing higher efficiency in real-time big data processing for Online Roadshows enables one to perform responsive decision-making and allows for more effective real-time targeting and overall trend forecasting (via historical data analysis) for an enhanced advertising experience. This is obviously very valuable for businesses, particularly in terms of digital marketing, as it could influence the planning of business strategies. This paper proposed a new dual-mode (real-time and historical) big data processing framework to allow the Online Roadshow to provide more responsive feedback and more efficient targeted advertising.

From the experimental results for the Historical Data Processing scenario, it was revealed that the proposed framework achieved a lower execution time for processing historical data at higher throughputs (0.1468 s). It also demonstrated moderate CPU usage

(ranging from 2.4% to 25.6%) but higher memory consumption amidst higher throughputs. Overall, the proposed framework is the most time-efficient.

On the other side, the performance analysis for the Real-time Data Processing scenario revealed that the proposed framework exhibited a lower execution time, ranging from 0.63 s to 6.69 s at lower throughputs (100, 300, and 500). It demonstrated efficient CPU usage, with the lowest usage being at the highest throughput of 2567 (10.15%). Additionally, the proposed framework showcased lower memory usage, ranging from 664 MB to 958 MB, compared to most combinations.

Looking towards the future, there is potential to expand the framework's capabilities so that it can support more formats and handle a wider range of data types generated in the Online Roadshow, including raw media data such as image, video, and audio data. This would involve enhancing the framework in terms of its functionality to process and analyze the different data formats, allowing for more complex personal audience preferences to be recognized and a more sophisticated level of personalization within the Online Roadshow.

Moreover, addressing security and privacy concerns related to engagement data is essential. Implementing robust measures to ensure data confidentiality and integrity, such as encryption, access controls, and secure storage mechanisms, should be a priority. This safeguards the sensitive information of the participants and promotes trust in the platform. Therefore, future work may include integrating these security and privacy concerns into the framework's data processing capabilities.

**Author Contributions:** Funding acquisition, M.-C.L.; Investigation, K.-R.L.; Project administration, L.-Y.O.; Supervision, M.-C.L.; Visualization, K.-R.L.; Writing—Original draft, K.-R.L.; Writing— Review and editing, M.-C.L. and L.-Y.O. All authors have read and agreed to the published version of the manuscript.

**Funding:** This research was funded by Telekom Malaysia Research and Development, RDTC/221073 and Multimedia University IR Fund, MMUI/220095.

**Data Availability Statement:** Not applicable. The study does not report any data.

**Conflicts of Interest:** The authors declare no conflict of interest.

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
