# Peer review of "A New Big Data Processing Framework for the Online Roadshow"

_2504-2289, doi:10.3390/bdcc7030123_

Round 1

Reviewer 1 Report

A New Big Data Processing Framework for Online Roadshow

1.       Big Data, Machine Learning, Data Science, AI, Deep Learning, etc. Big data will continue to improve models and progress research in many disciplines since they require data.

2.       There is no doubt about it that paper is timely;  but please highlight few more points on the theoretical contribution of this study and also the background of this study.

3.       I want authors to provide more literature support. The research gap should be emphasized.

4.       What new insights this manuscript adds to the existing literature?

5.       Can you please tell me Potential limitations ?

A New Big Data Processing Framework for Online Roadshow

1.       Big Data, Machine Learning, Data Science, AI, Deep Learning, etc. Big data will continue to improve models and progress research in many disciplines since they require data.

2.       There is no doubt about it that paper is timely;  but please highlight few more points on the theoretical contribution of this study and also the background of this study.

3.       I want authors to provide more literature support. The research gap should be emphasized.

4.       What new insights this manuscript adds to the existing literature?

5.       Can you please tell me Potential limitations ?

Author Response

Thank you so much for the review on this paper :)  

Please see the attachment for revision/feedback note.

Reviewer 2 Report

In this paper, the authors propose  a big data processing framework  to provide a better approach for historical and real-time big data processing for the engagement data in the Online Roadshow to provide higher data processing capability than conventional traditional programming model. The proposed framework allows campaign owner to process data in dual-mode to provide a better approach to analyze the overall trend and personalized engagement characteristics for each participant.

However, there are the following comments to the article:

1.       It is necessary to clearly highlight scientific novelty in the introduction and in the conclusions. The problem is that the work contains a lot of information that is common knowledge and the combination of software is not enough scientific novelty.

2.       The numbering of references to publications is presented chaotically, it is worth structuring the order of references in the text

3.       The results of research should be presented in quantitative form in the conclusions.

4.       It is necessary to more clearly describe the datasets on which the experiments were performed.

Author Response

(The authors gave the same response as above.)

Reviewer 3 Report

Dear authors, 

  thanks a lot for this great job. I have just few questions/suggestions to improve the quality paper: 

- check grammar and syntax, for sure I detected some typo, but better to pass with some specific tool; 

- Please elicit the acronym the first time they occours (just a check in the text) and even if possible put also some reference; 

- check the tables and improve padding of cells (in some case is not so readable); 

- the pseudo code in the figures, in my opinion, is not useful. Can be more effective some UML diagrams which allow you to present some static or dynamic view; 

- the experiment was conducted on a stand alone device (not so powerful). Why not a server, since we are speaking about web applications? Woule be more significant; 

- in the section result there is comparison between "'proposed framework" and other. I would suggest to put here also the configuration of your proposed framework. The cite [2] provide more details but something can be reported here to underatand differences with other frameworks. 

Thanks  a lot. 

Author Response

Thank you so much for the review on this paper :)

Reviewer 3

Item

Revision/Feedback

1. check grammar and syntax, for sure I detected some typo, but better to pass with some specific tool; 

A thorough check and revision are performed on the sentences and grammatical correctness with a few rounds of ProWritingAid checking. The paper is now grammatically sound and achieves good scores in a few aspects.

The ProWritingAid report is attached in the attachment field for reference.  

2. Please elicit the acronym the first time they occours (just a check in the text) and even if possible put also some reference; 

Changes made:

Line 41

I/O (input/output)

Line 69

application programming interface (API)

Line 83

Complex Event Processing (CEP)

Line 216

Graphic User Interface (GUI)

Line 231

Hypertext Transfer Protocol (HTTP)

Extensible Markup Language (XML)

3. check the tables and improve padding of cells (in some case is not so readable); 

The tables are checked and there seems to be no illegible parts as suggested in the comment. It is a known issue that some formatting may change in some version of MS Word. I have attached a PDF in the email as reference of the view on my side.

4. the pseudo code in the figures, in my opinion, is not useful. Can be more effective some UML diagrams which allow you to present some static or dynamic view; 

After careful deliberation, the pseudocodes are kept as they are quite clear and expressive in explaining the algorithms. This achieves our intended meaning.

5.  the experiment was conducted on a stand-alone device (not so powerful). Why not a server, since we are speaking about web applications? Woule be more significant; 

The proposed framework is indeed intended to be deployed on general computing devices for practical and more accessible usability. Hence, the experiments are kept to be using the standalone devices to reflect this goal.

6. in the section result there is comparison between "'proposed framework" and other. I would suggest to put here also the configuration of your proposed framework. The cite [2] provide more details but something can be reported here to underatand differences with other frameworks. 

Changes made :

Line 39 – 44

The Online Roadshow generates massive amounts of user preference data during the engagement duration with participants. The varieties of interactive game data involve data on body movement, voice input and I/O (input/output) interfaces (e.g., keyboard and mouse) [3]. These data are in the mixture of semi-structured and unstructured format (e.g., semi-structured game details and unstructured bodily movement coordination information). This makes it more challenging for traditional programming approaches to process the data.

Round 2

Reviewer 2 Report

The author took into account the previous comments. Additionally, provide an example dataset for greater clarity.

Author Response

Appreciate you responsive feedback :) Have a good day ahead !
